# Distinct dynamic connectivity profiles promote enhanced conscious perception of auditory stimuli
Başak Türker [1] ✉, Dragana Manasova[1,2], Benoît Béranger[1], Lionel Naccache[1], Claire Sergent [2,3] & Jacobo D. Sitt [1] ✉

The neuroscience of consciousness aims to identify neural markers that distinguish brain dynamics in healthy individuals from those in unconscious conditions. Recent research has revealed that specific brain connectivity patterns correlate with conscious states and diminish with loss of consciousness. However, the contribution of these patterns to shaping conscious processing remains unclear. Our study investigates the functional significance of these neural dynamics by examining their impact on participants' ability to process external information during wakefulness. Using fMRI recordings during an auditory detection task and rest, we show that ongoing dynamics are underpinned by brain patterns consistent with those identified in previous research. Detection of auditory stimuli at threshold is specifically improved when the connectivity pattern at stimulus presentation corresponds to patterns characteristic of conscious states. Conversely, the occurrence of these conscious state-associated patterns increases after detection, indicating a mutual influence between ongoing brain dynamics and conscious perception. Our findings suggest that certain brain configurations are more favorable to the conscious processing of external stimuli. Targeting these favorable patterns in patients with consciousness disorders may help identify windows of greater receptivity to the external world, guiding personalized treatments.

Our interactions with the environment are determined by an interplay between endogenous ongoing neural activity and our neural responses to external stimuli. Each moment, our brains process and integrate a wide variety of internal and external stimuli of different modalities. While some of these stimuli are processed consciously and contribute to our subjective experiences, most remain unconscious[1,2]. Neural events correlating with conscious perception are widely investigated in the literature, mainly by comparing the conscious and unconscious perception of the same stimulus using various paradigms such as masking[3], threshold stimuli presentation[4], and attentional blink[5]. These studies have shown that the same stimulus with a fixed intensity can induce different brain responses and subjective experiences.

One of the prerequisites for conscious perception is to be in a *conscious state*, such as wakefulness (high arousal and high awareness), as opposed to unconscious states, such as under anesthesia or coma (no arousal, no awareness), or disorders of consciousness (arousal with limited awareness). Recent studies in anesthetized non-human primates[6,7], and conscious and unconscious humans[8] have explored the time-varying dynamics of resting state functional connectivity. Unlike traditional resting-state connectivity analyses that utilize the entire scan[9–12], these studies have enabled the identification of recurring brain patterns that vary on a scale of seconds. They revealed distinctive connectivity patterns associated with different states of consciousness. Notably, certain brain patterns with long-range connectivity and negative interactions appear to be characteristic of a conscious state and diminish with the loss of consciousness. Additionally, thalamic deep brain stimulation that aimed at restoring consciousness in anesthetized non-human primates has been found to restore the afore-mentioned connectivity patterns[13]. However, the functional role of these brain patterns in conscious processing and the formation of subjective experience remains unknown. In this study, we investigate whether and how ongoing connectivity patterns influence the processing of external information, allowing it to become conscious or not.

The effect of spontaneous baseline brain activity fluctuations on perceptual outcome has been previously explored in different domains by contrasting perceived and unperceived trials. Electrophysiological recordings have shown that the pre-stimulus phase[14] and power[15,16] of alpha

[1]Sorbonne Université, Institut du Cerveau—Paris Brain Institute—ICM, Inserm, CNRS, Paris, 75013, France. [2]Université Paris Cité, Paris, 75006, France. [3]Integrative Neuroscience and Cognition Center—INCC, UMR 8002, Paris, 75006, France. ✉e-mail: basak.turker@icm-institute.org; jacobo.sitt@inserm.fr

activity, as well as the phase[17] and dynamics[18] of (infra-) slow cortical oscillations in the task specific regions, correlate with the perceptual outcome on trial-by-trial basis. Functional magnetic resonance imaging (fMRI) studies have found that cue-induced pre-stimulus activity reflects attentional allocation[19] and task preparation[20], and predicts task performance. Moreover, behavioral performance in Stroop[21] and motion discrimination tasks[22] as well as the perceptual outcome of ambiguous vase/face stimuli presentation[23] seem to vary depending on prior activity fluctuations in task-specific regions such as color-sensitive visual areas, motion-sensitive middle temporal region, and fusiform face area, respectively. In the nociception domain, pre-stimulus brain activity in the default-mode and fronto-parietal networks[24], along with the functional connectivity between brain areas involved in pain perception[25], are correlated with the subsequent perception of pain. And finally, baseline activations in sensory and attentional areas[26] and functional connectivity between different brain regions[27] have shown to predict perceptual performance in an auditory threshold stimulus detection task.

These studies consistently suggest that fluctuations in baseline brain activity can significantly impact our conscious perception of the external world. Yet, most research has primarily focused on individual activations within specific brain regions associated with a particular task. It's crucial to recognize that cognitive processes transcend localized regions and manifest through the coordination of various brain networks that process and exchange information. While some studies have explored the impact of pre-stimulus functional connectivity on perceptual outcome[25,27], they often focus on pairs of regions rather than considering the overall functional configuration of the brain. Although this approach provides valuable insights into networks relevant to the ongoing perceptual task, it falls short of offering a comprehensive description of the overall brain states that underpin conscious processing.

The primary objective of the current study is to examine how ongoing brain connectivity states influence the formation of conscious experience, specifically by affecting the ability to process external information. Our study encompasses several goals: (i) describing brain states as global connectivity configuration patterns involving different networks simultaneously, (ii) confirming the existence and characteristics of recurrent brain patterns observed during resting state[8], (iii) investigating the dynamics of brain states in a time-resolved manner, and (iv) exploring how brain

patterns associated with conscious states[8] affect the capacity for conscious perception. Using fMRI acquisitions, we showed that participants were more likely to detect auditory threshold stimuli when they exhibited connectivity patterns typical of conscious states[8]. Additionally, we observed a higher occurrence of these favorable connectivity patterns following stimulus detection, with participants more likely to either maintain or transition to these patterns after conscious perception. Our findings suggest that ongoing brain dynamics and conscious perception have a reciprocal influence on each other and that certain brain configurations provide a window of higher receptivity to the external world.

## Results

We investigated how ongoing brain configurations emerging from the coordination of different brain regions influenced the perception of external stimuli. 25 participants underwent fMRI recordings during an auditory detection task adapted from Sergent et al.[28] (Fig. 1). The task involved listening to the French vowel (/a/) that was embedded in continuous noise at 3 different signal-to-noise ratios (SNR -11, SNR -9, and SNR -7) around the detection threshold. The general sound level was adjusted for each participant via a staircase procedure prior to the task to ensure that they could detect a stimulus with an SNR of −9 dB in 50% of the trials. They also underwent a resting-state scan before the task.

### Ongoing brain dynamics are supported by brain patterns consistent with those identified in previous research

Applying the Hilbert transform and k-means clustering, we computed whole-brain connectivity patterns for each fMRI volume acquired during the resting-state scan (Fig. 2a). Our clustering procedure resulted in five distinct connectivity patterns (Fig. 2b). Subsequently, we utilized the centroids of these clusters to label the task data: each task fMRI volume was assigned to the nearest cluster based on its proximity to the cluster centroids. Upon visual inspection, our study's cluster centroids closely matched those documented by Demertzi et al.[8]. Patterns 1, 2, 3, and 5 in our study exhibited a robust similarity in their coherence profiles to those identified in the earlier study (Fig. 2c). Importantly, Pattern 1 corresponded to the brain state associated with conscious states and diminishing with loss of consciousness[8]. On the other hand, Pattern 4 was a new cluster, featuring a unique coherence profile not observed in the previous study. For a more

**Fig. 1 | Experimental procedure.** The experimental session began with a one-up-one-down staircase procedure which allowed participants to become familiar with the task and to adjust the stimulus volume such that a stimulus with an SNR of -9 would be detected in 50% of the trials. Following a 10minute resting state acquisition, participants completed an auditory detection task, during which they heard stimuli embedded in a continuous noise at different SNRs (−7, −9 and −11). They were instructed to press a button if they detected a stimulus. Stimuli were delivered randomly every 14 s (+/−1 s). At the end of each experimental block, participants verbally indicated on a scale of 7: (i) how tired they felt, (ii) how successful they think they were during the block, and (iii) their attentional focus during the block (1 for complete mind-wandering and 7 for complete focus on the task).

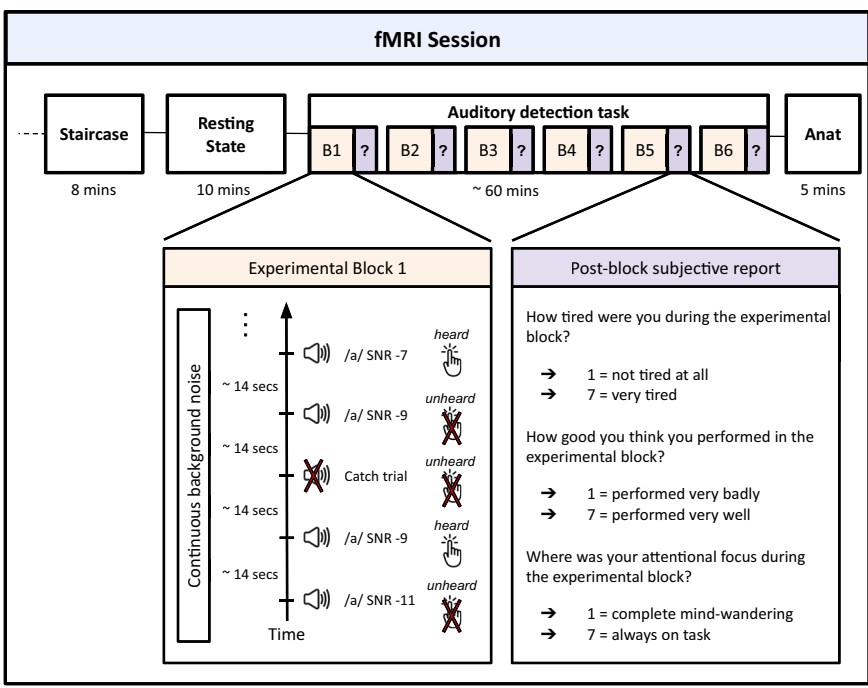

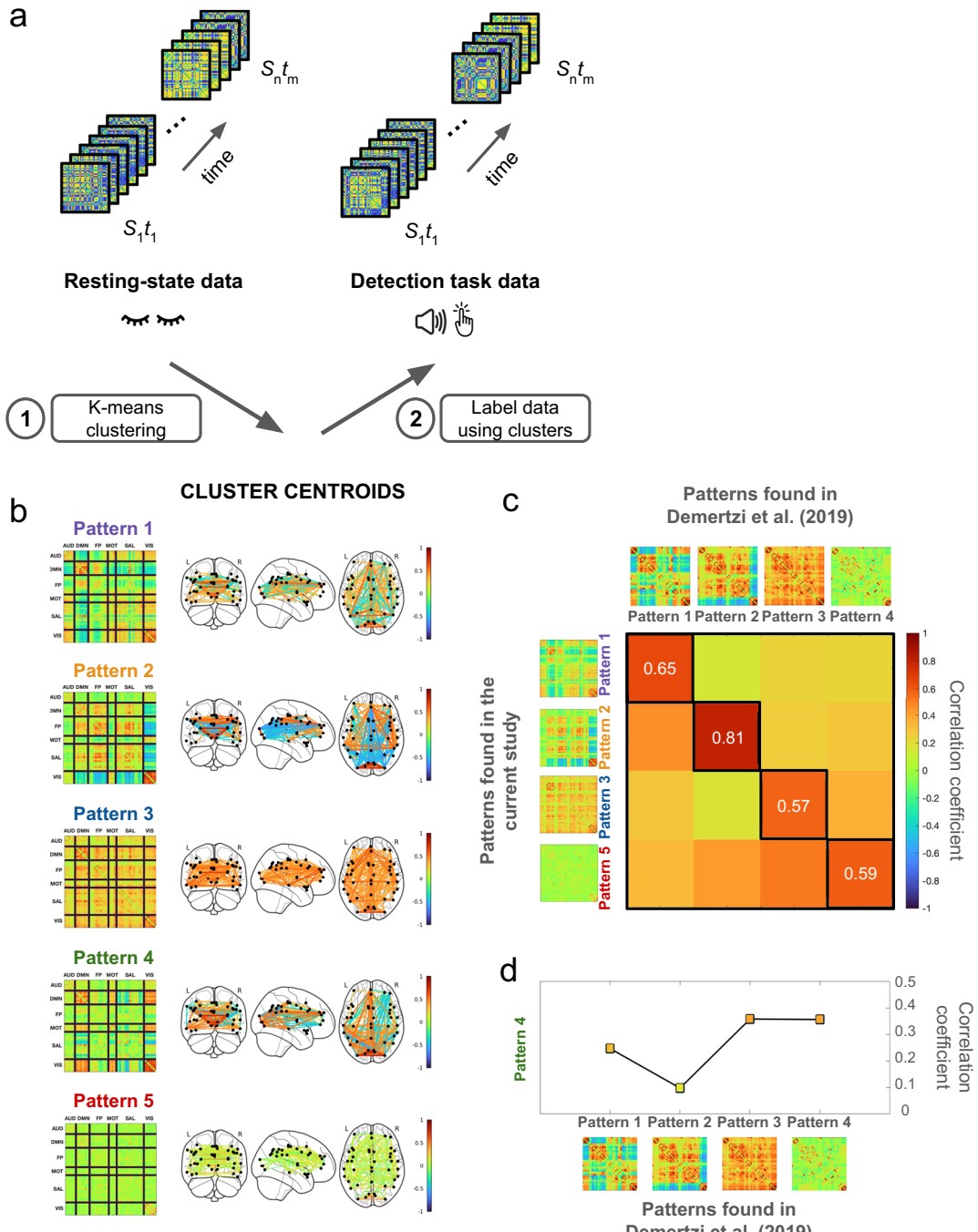

**Fig. 2 | Clustering procedure and the cluster centroids. a** Inter-areal connectivity matrices are computed for each resting state and task fMRI volume. Resting state connectivity matrices are divided into five distinct clusters using k-means clustering (1). The found cluster centroids are then used to classify the task data (2). By calculating the distance between each connectivity matrix from the task and the cluster centroids from the resting state data, the connectivity matrices from the auditory detection task are assigned to the closest of the five clusters. **b** Cluster centroids. **c** Comparison of the matching patterns. Pearson correlation between the coherence values of the patterns revealed a strong correspondence between the patterns of the two studies. **d** Formal comparison of Pattern 4 to the patterns found in Demertzi et al. This newly discovered pattern displayed a lower degree of similarity to the patterns identified in the earlier study.

formal comparison, we calculated Pearson correlation coefficients between the coherence values of the patterns from the two studies (Fig. 2c). Our analysis revealed very strong correlations between our patterns and those from the earlier study, showcasing a one-to-one matching (Pattern 1: $rho = 0.65$, $p < 0.0001$; Pattern 2: $rho = 0.81$, $p < 0.0001$; Pattern 3: $rho = 0.57$, $p < 0.0001$; Pattern 5: $rho = 0.59$, $p < 0.0001$). As visually depicted, Pattern 4 (the new pattern) demonstrated lower similarity to the patterns from Demertzi et al., with correlation values ranging from 0.099 to 0.36 (Fig. 2d).

The clustering procedure successfully yielded well-defined cluster centroids (Patterns), showcasing functional connections that respected network borders despite the sparse nature of the connectivity configurations in the input data (Supplementary Fig. 1a). This indicated the capability of our clustering procedure to identify commonalities among the connectivity patterns. To ensure that the similarity between our cluster centroids and those from the prior research was not merely a feature inherent to the method employed, we conducted a control analysis. We generated a surrogate dataset and repeated the phase-based connectivity analyses along

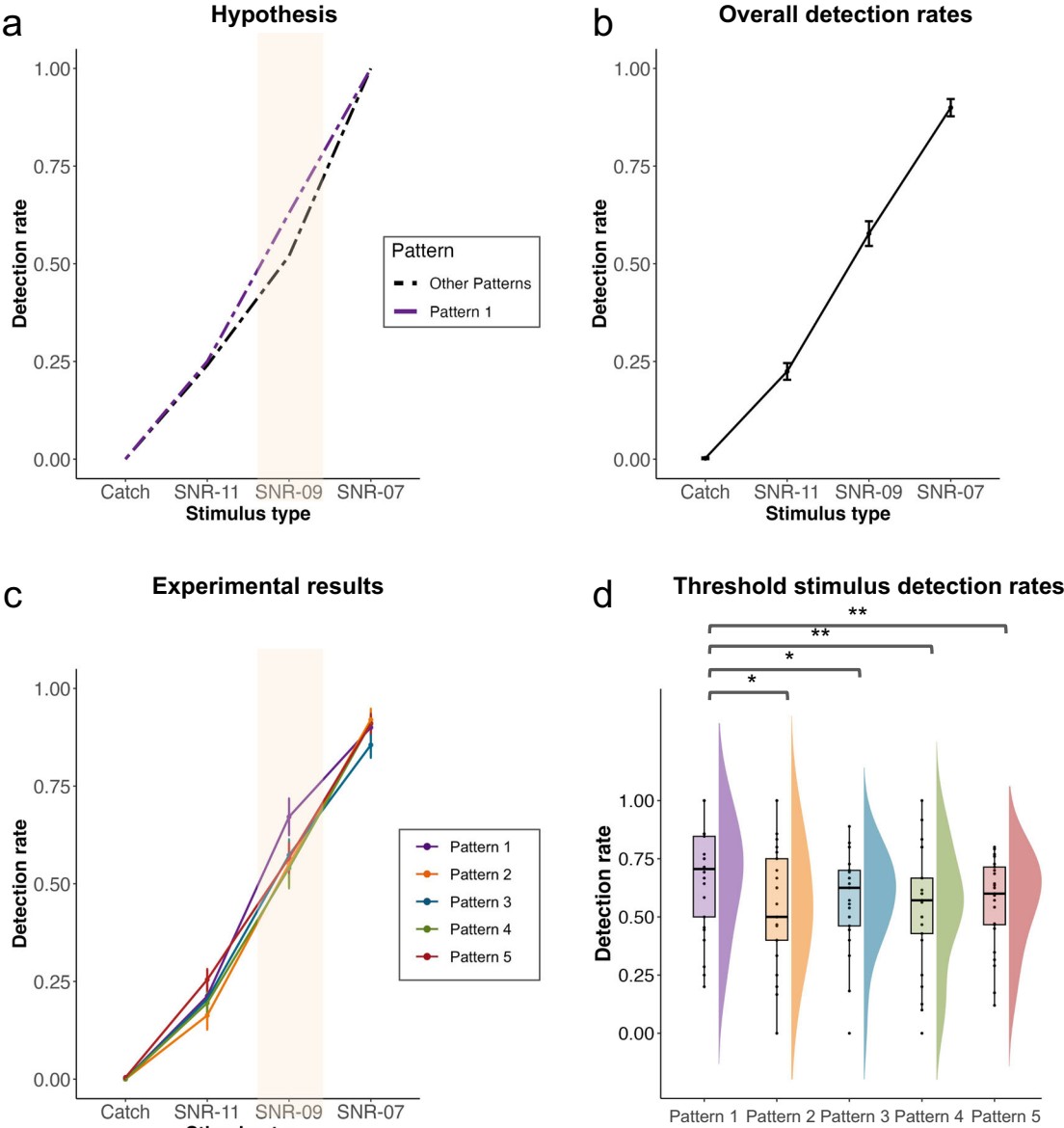

**Fig. 3 | Detection rates vary with SNRs and pre-stimulus connectivity configurations. a** Hypothesis of the experiment. We predicted increased detection rates of the threshold stimulus (SNR -9) when the consciousness-related brain pattern (Pattern 1) was present at the moment of stimulation. **b** Overall detection rates for different SNR. **c** Participant-averaged psychometric curves for each pattern. Error bars indicate the standard error. Participants were more likely to detect the threshold stimulus when displaying Pattern 1 compared to the other patterns. **d** Detection rates of the threshold stimulus SNR-09 for different patterns. Pattern 1 was associated with increased detection compared to the other patterns. No difference was found between Patterns 2–5. Each point represents one participant ($n = 25$), and the center lines of the box plots represent medians.

with the clustering procedure. By randomly shifting the time-series from each ROI and participant, we disrupted the temporal relationship between different ROIs while maintaining the temporal order within each time series. As anticipated, clustering procedure applied to surrogate data resulted in indistinguishable centroids lacking inter-areal connectivity (Supplementary Fig. 1b). The absence of informative clusters in the surrogate analysis demonstrated that the patterns identified in the experimental data truly reflected the brain's connectivity state and were not artifacts of the methodology.

## Certain connectivity profiles are associated with enhanced conscious perception

We focused on the task data and hypothesized that the detection of the threshold stimuli would vary depending on the connectivity pattern present at the time of presentation. More precisely, we predicted that participants

would be more likely to detect the threshold stimuli if they had the connectivity pattern (Pattern 1) which was previously shown to be the most typical of conscious states[8] (Fig. 3a). Additionally, we considered whether the perception of a previous stimulus could influence the perception of the current one. To account for this potential effect, we tested both hypotheses with a linear mixed model with subject ID as a random effect and pattern ID, SNR and previous stimulus detection status (detected vs. undetected) as fixed effects.

Overall detection rates (all patterns considered) were 0.90 for SNR -7, 0.58 for SNR -9, 0.22 for SNR -11 and 0.002 for catch trials (Fig. 3b). As hypothesized, we found a significant interaction between the SNR and Pattern ID ($\chi^2(12) = 22.67$, $p = 0.031$). Detection rates were significantly higher when participants were presenting Pattern 1 (mean = 0.67; median = 0.71) compared to Pattern 2 (mean = 0.55; median = 0.50; $t = 2.69$; $p = 0.019$ after FDR correction; Cohen's $d = 0.28$), Pattern 3 (mean = 0.57;

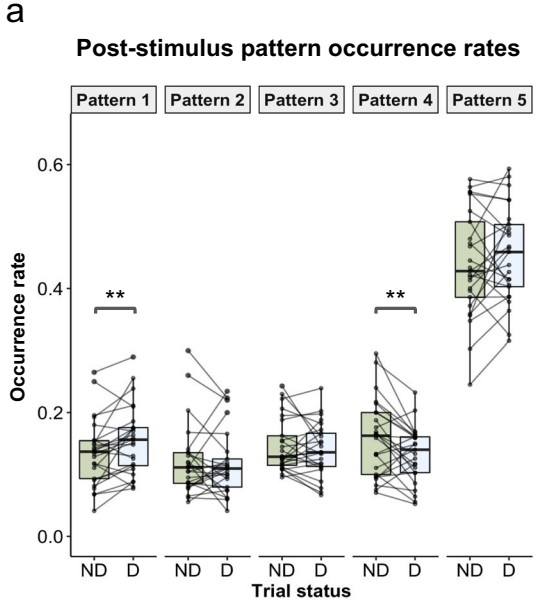

**a**

**Post-stimulus pattern occurrence rates**

**Fig. 4 | Conscious perception of threshold stimuli impacts subsequent brain dynamics. a** Subject-averaged pattern probabilities within the 9 s post-stimulus time window following a detected (D) and undetected (ND) threshold stimulus (SNR-9) presentation. The post-stimulus probabilities of Pattern 1 and Pattern 4 varied depending on whether the stimulus was detected or undetected. Each point represents one participant ($n = 25$), and the center lines of the box plots represent medians. **b** Time-resolved pattern probability analysis. Subject-averaged pattern probabilities across detected (blue) and undetected (green) trials in a time window spanning from 3 s before the presentation of the threshold stimulus to 9 s after stimulation. Shaded areas represent standard errors. Pattern 1 showed higher occurrence in detected trials ($V = 353$, $p = 0.042$, marked with a red line) at the moment of stimulus presentation ($t = 0$). It also exhibited a trend of higher occurrence in detected trials in the post-stimulus period, particularly 4 s after the stimulation ($V = 254$, $p = 0.086$, marked with an orange dashed line). Conversely, Pattern 4 displayed a trend of lower occurrence in detected trials 5 s following the stimulus presentation ($V = 58$, $p = 0.067$, marked with orange dashed line).

median = 0.62; $t = 2.67$; $p = 0.019$; $d = 0.26$), Pattern 4 (mean = 0.54; median = 0.57; $t = 3.49$; $p = 0.0042$; $d = 0.35$) and Pattern 5 (mean = 0.57; median = 0.60; $t = 3.34$; $p = 0.0042$; $d = 0.27$) only for the threshold SNR -9 (Fig. 3c–d). No significant differences were found in the other SNRs except between Pattern 2 and Pattern 5 at SNR -11 (mean: 0.16 vs. 0.25; $t = -3.29$; $p = 0.01$; $d = -0.29$). When comparing the detection rates at SNR-09 for each pattern to the overall detection rate resulting from the staircase procedure (0.58), we found that only Pattern 1 had significantly higher detection rates (one-sided Wilcoxon signed-rank test, $V = 231$, $p = 0.033$). The detection rates for the other patterns did not differ from the overall detection performance (Pattern 2: $V = 148$, $p = 0.66$; Pattern 3: $V = 174$, $p = 0.38$; Pattern 4: $V = 147$, $p = 0.66$; Pattern 5: $V = 170$, $p = 0.42$). Thus, Pattern 1 was uniquely associated with higher detection rates, while the other patterns did not exhibit this effect.

We also observed a main effect of previous detection ($\chi^2(1) = 21.20$, $p < 0.0001$), indicating an increased likelihood of detection if the previous stimulus was also detected. This effect was observed regardless of the pattern ($\chi^2(3) = 6.59$, $p = 0.086$, only a tendency was found) and the SNR ($\chi^2(4) = 7.43$, $p = 0.11$). Importantly, our key findings were not solely dependent on including this covariate in our model. Even without considering previous detection status, a significant pattern*SNR interaction persisted ($\chi^2(12) = 22.27$, $p = 0.035$). Detection rates were significantly higher when participants were presented with Pattern 1 compared to Pattern 2 ($t = 2.66$; $p = 0.021$; $d = 0.27$), Pattern 3 ($t = 2.63$; $p = 0.021$; $d = 0.25$), Pattern 4 ($t = 3.42$; $p = 0.0053$; $d = 0.34$) and Pattern 5 ($t = 3.28$; $p = 0.0053$; $d = 0.26$) only for the threshold SNR -9, confirming our initial results.

Next, we assessed whether the reaction time differed depending on the ongoing patterns. Although we did not find any significant differences between patterns, we found a main effect of the SNR ($\chi^2(2) = 276.83$, $p < 0.0001$). Not surprisingly, the reaction times were faster for SNR -7 (mean = 1.02 s) compared to SNR -9 (mean = 1.10 s; $t = -9.43$; $p < 0.0001$; $d = -0.48$) and SNR -11 (mean = 1.20 s; $t = -13.16$; $p < 0.0001$; $d = -1.00$).

## Stimulus detection increases the occurrence of patterns favorable for conscious perception

To further investigate the dynamic interplay between the ongoing brain patterns and conscious perception, we assessed whether conscious perception, in turn, alters ongoing brain configurations. To do so, we calculated the occurrence probability of each pattern in a 9 s window following the threshold (SNR -9) stimulus presentation and compared the subject-averaged probabilities between detected (D) and not detected (ND) trials using two-sided Wilcoxon signed-rank test. This time-window allowed us to have enough data points to compute post-stimulus pattern probabilities at the trial level without including patterns from the pre-stimulus period of the next trial. The occurrence of Pattern 1 increased following detection, with participants being more likely to transition to highly connected patterns after a stimulus detection ($V = 258$, $p = 0.0088$) (Fig. 4a). Conversely, we found an increased occurrence of Pattern 4 when participants did not detect the stimuli ($V = 69$, $p = 0.01$). The occurrence of the other patterns did not differ between detected and not detected trials (Pattern 2: $V = 147$, $p = 0.69$, Pattern 3: $V = 138$, $p = 0.52$, Pattern 5: $V = 204$, $p = 0.27$).

These results indicated an overall fluctuation in the occurrence of Patterns 1 and 4 depending on whether participants consciously perceived the stimulus. We performed a time-resolved analysis to further explore the exact timing of these changes. We extracted pattern information within a time window spanning from 3 s prior to the presentation of the threshold stimulus (pre-stimulus period) to 9 s after stimulation (post-stimulus period) for each trial. Subsequently, we computed, for a given time point, the probability of pattern occurrence across trials where the stimulus was detected and those where it went undetected. Subject-wise comparison of average pattern probabilities between detected and undetected trials revealed no significant differences in occurrence probabilities across conditions during the pre-stimulus period (Fig. 4b). Conversely, during stimulus presentation (time = 0), the occurrence probability of Pattern 1 was statistically higher in detected trials compared to undetected trials ($V = 353$,

$p = 0.0034$; after FDR correction $p = 0.042$), corroborating our initial findings. In the post-stimulus period, the probability values of Pattern 1 remained consistently higher in the detected trials, reaching statistical significance 4 s following the stimulation ($V = 254$, $p = 0.013$; a trend after FDR correction $p = 0.086$). For Pattern 4, a difference between detected and undetected trials emerged 5 s after the stimulus presentation ($V = 58$, $p = 0.0051$, a trend after FDR correction $p = 0.067$). A difference was also noted for Pattern 3 2 s after stimulus presentation, although it did not withstand multiple comparisons ($V = 72$, $p = 0.015$, not significant after FDR correction $p = 0.20$).

Finally, we leveraged post-block subjective ratings to delve into the relationship between the occurrence of the patterns and participants' subjective mental states within a given block (Supplementary Text). We found a significant correlation between feelings of tiredness and the occurrence of Pattern 4 ($rho = 0.3$, $p = 0.004$, after FDR correction), indicating that participants more frequently exhibited Pattern 4 in blocks where they reported feeling tired (Supplementary Fig. 2).

Altogether, our results suggest that ongoing whole-brain functional connectivity influences our capacity to consciously perceive external stimuli. Certain whole-brain connectivity profiles that were previously associated with conscious states[8] consistently enhanced detection of threshold auditory stimuli. Moreover, the conscious perception of the stimulus also influenced the subsequent brain dynamics. Indeed, occurrence probabilities of these 'favorable' patterns increased following conscious perception, underscoring a reciprocal relationship between ongoing brain dynamics and conscious perception. Our findings demonstrate that processing of external stimuli may vary depending on the specific brain state we are in and that certain brain states promote greater susceptibility to the external world.

## Discussion

Our study provides compelling evidence that ongoing brain connectivity profile influences our ability to process external stimuli. Participants were more successful in detecting auditory stimuli at threshold when they exhibited brain patterns characteristic of conscious states[8] during the stimulation. Previous research has shown that while such patterns frequently expressed in healthy individuals, they diminish with loss of consciousness[8]. Our results suggest that these patterns facilitate external information processing, which could explain why they are rarely observed in vegetative state patients who lack signs of external awareness. Conversely, these patterns become more frequent in patients transitioning to a minimally conscious state, where they exhibit transient moments of external awareness.

One could argue that the pattern observed during stimulus presentation might be influenced by residual activity from the previous response. To mitigate this potential influence, we implemented unusually long inter-trial intervals (14 s +/−1 s) to minimize carryover effects from the previous trial onto the current one. In our analysis of post-detection occurrence rates, we restricted our focus to a 9-s time window rather than the full 14 s post-stimulus period. This ensured we had enough data points to calculate post-stimulus pattern probabilities at the trial level while avoiding the inclusion of patterns from the pre-stimulus period of the next trial. This strategy helped us avoid the potential introduction of confounding factors from overlapping time points in both analyses.

Given the hemodynamic response latencies, brain pattern observed at the moment of stimulation (time 0) reflects brain activity from several seconds earlier. We chose to focus on this time point rather than 4–5 s post-stimulation (which would capture the peak of hemodynamic activity present at the stimulus presentation) because post-stimulus time points would also include activity induced by the stimulus and its detection. This would result in a mixture of baseline brain activity and stimulus/response-induced activity, making it difficult to disentangle their contributions. To avoid this, we focused on the moment of stimulus presentation, reflecting purely ongoing baseline brain activity unaffected by the stimulation. We chose time 0 over earlier time points (e.g., time -1 or -2) to avoid a larger time lag. Our extended inter-trial intervals allowed hemodynamic responses from the previous trial to return to baseline, leaving us with spontaneous fluctuations

in ongoing brain activity similar to a resting state. Given the technical limitations of fMRI (i.e. poor time resolution), this approach was optimal for investigating the effect of baseline brain state on conscious content. Focusing on later time points would have contaminated our signal with stimulus-induced activity, thus not purely reflecting baseline brain activity.

It's worth noting that the increase in performance was only observed for stimuli at threshold; the detection rates for supra- and sub-threshold stimuli remained unaffected by the ongoing brain pattern. This may be because sub-threshold stimuli do not provide enough bottom-up information to trigger conscious processing[1] and supra-threshold stimuli are strong enough to provoke conscious processing regardless of the ongoing brain pattern. Therefore, threshold stimuli provide a sweet spot where changes in ongoing brain state can impact the processing of stimuli and give rise to conscious access when the brain state is favorable.

Previous studies using stimuli at threshold to examine the impact of baseline brain activity on conscious perception have mainly focused on variations in local activity within task-related regions[14,16,22,23,25]. While these localized activities can offer insights into the excitability of the task-related regions, establishing their connection to consciousness proves challenging. Some current theories of consciousness propose that perceptual awareness arises from long-range interactions between different brain regions[2,29]. To test the predictions of these prevailing theories, it becomes imperative to examine how the coordination of diverse brain networks impacts conscious perception. Our results align with the global theories of consciousness, as opposed to local theories of consciousness[30,31], by demonstrating that specific configurations of long-range connectivity are more favorable for conscious perception.

Recent research has revealed that the processing of high-level stimuli, such as audio-visual movies, synchronizes ongoing brain connectivity dynamics[32]. Our findings contribute to this body of literature by illustrating how the conscious perception of threshold stimuli influences subsequent brain patterns. Specifically, connectivity patterns conducive to conscious perception were more frequently observed following a successful detection. This suggests that once the brain consciously processes a stimulus, it tends to remain or transition into states that are favorable for conscious perception. This stability in the brain's receptiveness to the external world over a certain period of time aligns with the concept of perceptual hysteresis[33], which refers to the influence of an immediately preceding perception on the current one. Based on our findings, we could speculate that participants likely transitioned into or maintained these favorable states after a detection, thereby increasing their chances of detecting subsequent stimuli. Indeed, we observed that prior detections increased the likelihood of current detections, suggesting a hysteresis effect. Although the interaction with the pattern type was only marginally significant, we believe that all these findings reflect a common hysteresis phenomenon facilitated by the transition to favorable brain patterns. Further studies are required to explore and validate this interpretation thoroughly.

Another interesting finding was the higher occurrence of Pattern 4 following stimuli that participants failed to detect. This result aligns with subjective ratings of tiredness, showing a significant correlation with the occurrence of Pattern 4 in a given block. If this brain pattern is indeed associated with tiredness, it could elucidate the heightened occurrence of the pattern when participants were unable to detect the threshold stimulus, possibly due to fatigue. However, we did not observe a predictive effect of this pattern on subsequent detection, meaning that displaying such a pattern did not decrease detection rates. Further research is needed to clarify this aspect.

In a recent study, Mortaheb and colleagues investigated the relationship between ongoing brain connectivity and participants' ongoing mentation[34]. The authors found that mind blanking—a wake state without any mental content—was associated with brain patterns exhibiting positive connectivity among different brain regions, a pattern also observed in our study (Pattern 3) and in a prior study[8]. Previous research has linked mind blanking to sluggish responses in sustained attention tasks[35,36]. In light of this literature, we would have expected to observe longer reaction times when

participants exhibited this positively connected pattern, indicating potential mind-blanking. However, we did not find such an effect in our study. It is important to note some important differences between Mortaheb et al.'s study and ours. Unlike Mortaheb's study, our study wasn't specifically designed to explore mind blanking. We did not include any direct probes about mind blanking, but rather inquired about participants' mind wandering at the end of each block. Moreover, while reaction times can serve as an indirect measure of mind blanking, subjective reports offer more direct evidence. In the future, combining both our and Mortaheb's paradigms could offer a more comprehensive understanding of how ongoing brain and mental states interact with the ability to perceive the external world.

Our study takes a distinct approach from prior research in this field. While previous studies typically compared pre-stimulus brain activity in detected versus undetected trials[14–18,22–27,37], we opted for a hypothesis-driven approach by independently labeling brain patterns, irrespective of the trial outcome. Rather than looking for differences in brain activity between detected and undetected trials, we predicted and showed that certain pre-defined brain activity configurations can enhance conscious perception. This strategy allowed us to illustrate the relevance of specific brain patterns for both conscious access and conscious states, thereby bridging these two areas of research that are often explored in isolation.

Our study also replicated the brain connectivity patterns observed during rest in a prior publication[8]. Despite employing different MRI scanners and scanning different populations, our findings were consistent with the earlier study, suggesting that our method can produce reliable results despite variations in experimental conditions. Some might argue that the similarity between the two studies is due to an inherent feature of our method rather than to the experimental data. To rule out this possibility, we tested our clustering method on surrogate data with the same characteristics as our original dataset. This control analysis produced completely different connectivity patterns than the original, which lacked all types of connectivity, illustrating the robustness of our method in different experimental settings. One notable difference between the patterns of the two studies was that our inter-areal coherence values were lower than those in the prior study. This discrepancy might be attributed to the fact that we implemented additional denoising steps, such as regressing out heart and respiratory activity from the ROI timeseries. This extra step potentially eliminated the correlation between different regions induced by physiological artifacts, retaining only the 'real' coherence among brain regions and subsequently reducing the coherence values. However, the denoising procedure did not impact the coherence profiles.

Our research has opened up possibilities for other exciting studies that examine how fluctuations in the ability to perceive external information from different modalities occur. Given the significance of these brain configurations for both conscious access and conscious states, we believe that the observed results extend beyond the auditory domain alone. We anticipate that future studies will reveal that the perception of stimuli from other modalities, such as visual or somatosensory stimuli, is similarly influenced by these ongoing patterns.

The findings of our study indicate that ongoing patterns of brain connectivity, which are associated with conscious states, may actively contribute to shaping conscious experiences by altering the capacity to perceive external information. In the future, identification of these favorable patterns in real-time in individuals with consciousness disorders could enable us to target periods of increased receptivity to the external world, paving the way for the development of personalized patient-care protocols.

## Methods
### Participants
Twenty-six healthy participants were recruited for this study (13 women, mean age: 24.6 ± 4.2 years, 25 right-handed). All participants were native French speakers with good hearing and without any neurological or psychiatric disorders. They gave written consent prior to the experiment and were remunerated €70 for their participation in the study. One male participant was excluded from the study due to technical issues during the MRI

acquisition. The protocol had been approved by the local ethics committee (promoted by the INSERM, CPP Ile-de-France 6, C13-41). All ethical regulations relevant to human research participants were followed.

### Experimental design and procedure
Participants underwent fMRI recordings while performing an auditory detection task. They were asked to detect a French vowel (/a/) embedded in continuous noise[28], at 3 different signal-to-noise ratios (SNR -11, SNR -9 and SNR -7) around the detection threshold. Stimuli were delivered in a randomized fashion every 14 s (+/−1 s) using MRI-compatible headphones. Participants had their eyes closed in the fMRI scanner and pressed a button with their right thumb when detected a stimulus. The sound level was adjusted for each participant via a staircase procedure prior to the task to ensure 50% detection at SNR -9. This resulted in higher detection rates at SNR -7 and lower detection rates at SNR -11. The task consisted of 6 blocks of 8 min separated by a small rest period. Thirty stimuli were presented in each block (ten per SNR level) in addition to three catch trials. Thus, the whole task contained 198 trials (33 per block). After each block participants were asked to verbally indicate on a scale of 7: (i) how tired they felt, (ii) how successful they think they were during the block, and (iii) their attentional focus during the block (1 for complete mind-wandering and 7 for complete focus on the task). Participants also underwent a 10 min resting state with eyes closed before the task and a 5 min anatomical scan after the task. fMRI was acquired throughout the whole experimental session including the staircase procedure and the resting state. The total experimental session took ~2 h.

### MRI acquisition parameters
All MRI data were acquired on a 3 T Siemens Prisma System. T2*-weighted whole brain resting state images were acquired with a multi-band gradient-echo planar imaging (EPI) sequence (600 volumes, 48 slices, slice thickness: 3 mm, TR/TE: 1000 ms/30 ms, voxel size: 3 × 3 × 3 mm, multiband acceleration factor: 3, flip angle: 60°). Functional MRI images during the detection task were acquired using the same sequence. The cardiac and respiratory activities were also recorded during the fMRI acquisitions. An anatomical volume was acquired using a T1-weighted MPRAGE sequence in the same acquisition session (192 or 256 slices, slice thickness: 1 mm, TR/TE: 2.300 ms/ 2.76 ms, voxel size: 1 × 1 × 1 mm, flip angle: 9°).

### fMRI preprocessing
The raw MRI data underwent preprocessing and denoising using custom MATLAB (The MathWorks) scripts. The preprocessing included segmentation using CAT12[38], realignment, co-registration, and normalization into the MNI152 (Montreal Neurological Institute) space as implemented in SPM12[39]. We did not perform slice-timing correction as our TR was already short (1 s). We also avoided spatial smoothing of our data. The susceptibility-induced off-resonance field distortions were corrected using the *topup* procedure[40] as implemented in FSL[41], providing a more accurate representation of the brain. Peripheral physiological data recorded during the scans such as respiration and cardiac pulsation were extracted using PhysIO Toolbox[42]. White matter masks, realignment parameters as well as their first and second order derivatives, cardiac and respiratory signals were included as nuisance regressors in the generalized linear model (GLM) in order to denoise the data. Average time-series were extracted from 42 regions of interest (ROIs) defined as 5 mm radius spheres centered at specified MNI coordinates (as listed in Supplementary Table 1).

### Time-varying functional connectivity patterns
**Resting state**. After preprocessing, the extracted ROI time-series were converted into a complex representation using their original signal and the Hilbert transform. The instantaneous phase was calculated by taking the inverse tangent of the ratio of the imaginary and real parts and then by "wrapping" into the [-π,π] interval. This created a series of instantaneous phases for each ROI. Next, the phase differences between each ROI pair were determined at each time point using cosine similarity, allowing the

brain's connectivity configuration at each time point to be represented in an 861-dimensional space (each dimension represented the coherence of a pair of ROIs). Data from all participants were combined and k-means clustering (with k values of 3, 4, 5, 6, and 7) was applied with 1000 repetitions using the Manhattan distance to identify recurrent connectivity configurations. The silhouette method determined that five clusters provided the best classification. The connectivity configuration at each time point (a 42 by 42 phase coherence matrix) was then labeled with one of the 5 cluster centroids. Finally, the participants' brain activity during the scans was represented as a sequence of the five centroids.

**Detection task.** For each task fMRI volume, inter-areal coherence matrices were calculated using the Hilbert transform and cosine similarity as described above. These matrices were then assigned to the closest of the five clusters that were computed using the resting state data.

### Statistics and reproducibility

Statistical analyses were conducted in R[43] using lme4[44], emmeans[45], and car[46] packages. To account for multiple comparisons, all statistics were corrected using the False Discovery Rate (FDR) Benjamini-Hochberg procedure. Linear mixed models with subject ID as a random factor were used to investigate SNR and Pattern ID on detection rates and reaction times. The statistics for both detection rates and reaction times were calculated at the individual subject level, and the observations in the model were weighted based on the number of trials performed by each participant. Only responses with a latency between 400 ms and 2000ms were considered in the analyses. Importantly, an arcsine transformation was applied to the detection rates and an inverse transformation was applied to the reaction times (1/RT) to better meet the model assumptions. The assumptions of the linear models were assessed visually through residual distributions and Q-Q plots, and the significance of individual factors was evaluated using Wald $X$-tests. Subject-averaged post-stimulus pattern probabilities were compared between detected and undetected trials using paired Wilcoxon signed-rank tests. Finally, the relationship between pattern probabilities and subjective reports was assessed using Spearman correlations rather than Pearson correlations since they are more suited for ordinal scales such as subjective ratings.

### Reporting summary

Further information on research design is available in the Nature Portfolio Reporting Summary linked to this article.

## Data availability

All data that support the findings of the study can be found in OSF.

## Code availability

Custom analysis scripts can be found in OSF.

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

## Acknowledgements
This study was funded by the European Union (ERA PerMed JTC2019 "PerBrain", grant to JDS). BT received a PhD grant from French Ministry of Higher Education and Ecole Normale Supérieure. DM received individual funding from Ecole Doctorale Frontières de l'Innovation en Recherche et Education–Programme Bettencourt.

## Author contributions
B.T., C.S., and J.D.S. designed the research; B.T. and D.M. acquired the data; B.B. provided technical guidance for the acquisition and pre-processing of the data; B.T. and D.M. analyzed the data; B.T., D.M., L.N., C.S., and J.D.S interpreted the data. B.T. wrote the manuscript and all authors contributed to its editing.

## Competing interests
The authors declare no competing interests.
