## [Peer review file · Communications Biology]

Reviewers' comments:

Reviewer #1 (Remarks to the Author):

This study is a follow-up on an influential study published by this group a few years back (Demertzi et al., 2019) which previously identified patterns of brain activity associated with different states of consciousness. This new manuscript now aims to determine if those same patterns of brain activity can also be predictive of the content of consciousness in an auditory task. This is an important research question as the two fields of research tend to evolve in parallel with scarce attempts at connecting them.

The manuscript is generally clear, straightforward and convincing. My main comment is that it appears that the hemodynamic response function has not been accounted for in the interpretation of the results. For instance, what does it mean for pattern 1 to be more expressed at the onset of an auditory stimuli if this pattern likely reflects brain activity that occurred 4-5 seconds before that event? This is not so problematic considering that pattern 1 and 4 are also differentially expressed 4 seconds after stimulus onset. However, I am not sure how to interpret the results at time 0.

Minor comments:

- The rationale for including the previous stimulus detection status as a covariate is not so well explained. Please provide a bit more context in the manuscript.
- In the abstract, it is not so clear what "high pattern" refers to.
- Page 11 line 261: the word "stimuli" is missing.

Reviewer #2 (Remarks to the Author):

This is an elegant study using fMRI data during rest and an auditory detection task to characterise the effects of specific brain connectivity patterns on the likelihood of consciously perceiving external stimuli. The authors provide evidence that participants are more likely to detect threshold auditory stimuli when highly connected brain patterns are present and, in turn, conscious access increases the likelihood of a subsequent highly connected pattern appearing.

The study is well conducted and the results are convincing. I only have minor comments:

1. A successful detection increases the likelihood of also detecting the following stimulus and also is associated with the lingering of pattern 1. With these two results in mind, the fact that Pattern 1 is more likely to be present at the point that a new stimulus is detected could be explained by this lingering activity from the previous stimulus. In other words, pattern 1 could be associated with the brain's post-detection configuration rather than having a facilitatory effect per se. I believe this is partially addressed with the analyses displayed in Figure 4 B, that shows a reduction of the pattern occurrence after 9 seconds but I think it would be helpful to see an explicit acknowledgment of this and the authors thoughts about it in the discussion.
2. While Pattern 1 is significantly associated with higher detection rates, several of the other patterns appear to also have detection rates above chance (this can be seen e.g. in Figure 3). I like that the authors are careful in their wording when discussing their findings and talk about 'favourable / unfavourable patterns' but I think it'd be interesting to have some reflections in the discussion about what this means for our understanding of the relationship between these patterns and conscious / unconscious states.
3. Please spell check the first paragraph of the discussion. There is an unfinished sentence and a typo.

Response to reviewers

We thank the two reviewers and the editor for their time and effort reviewing our manuscript. Below, we provide a point-by-point response to the issues raised by the reviewers. The original reviewer's comments are shown in black and our responses in blue. All page numbers refer to the revised manuscript file with tracked changes in red.

Reviewer #1:

This study is a follow-up on an influential study published by this group a few years back (Demertzi et al., 2019) which previously identified patterns of brain activity associated with different states of consciousness. This new manuscript now aims to determine if those same patterns of brain activity can also be predictive of the content of consciousness in an auditory task. This is an important research question as the two fields of research tend to evolve in parallel with scarce attempts at connecting them.

The manuscript is generally clear, straightforward and convincing.

We thank the reviewer for their encouraging words! We are glad that the significance of our work for consciousness research is appreciated.

My main comment is that it appears that the hemodynamic response function has not been accounted for in the interpretation of the results. For instance, what does it mean for pattern 1 to be more expressed at the onset of an auditory stimuli if this pattern likely reflects brain activity that occurred 4-5 seconds before that event? This is not so problematic considering that pattern 1 and 4 are also differentially expressed 4 seconds after stimulus onset. However, I am not sure how to interpret the results at time 0.

The reviewer rightly points out that the brain pattern observed at the moment of stimulation (time 0) actually reflects brain activity from several seconds earlier. We chose to focus on this time point rather than 4-5 seconds post-stimulation (which would capture the peak of hemodynamic activity present at the stimulus presentation) because post-stimulus time points would also include activity induced by the stimulus and possibly its detection. This would result in a mixture of baseline brain activity and stimulus/response-induced activity, making it difficult to disentangle their contributions. To avoid this issue, we focused on the moment of stimulus presentation, reflecting purely ongoing baseline brain activity unaffected by the stimulation.

We also chose the moment of stimulus presentation (time 0) over prior time points (e.g., time -1 or -2) because those would reflect brain activity from even further before the stimulation, resulting in a larger time lag.

To ensure the observed patterns at stimulus presentation were not influenced by the previous trial (stimulus and response), we used unusually long inter-trial intervals of 14 seconds (+/- 1 second). This allowed hemodynamic responses from the previous trial to return to baseline, leaving us with spontaneous fluctuations in ongoing brain activity similar to a resting state.

We acknowledge that this strategy is not ideal, as the brain patterns at time 0 do not actually reflect brain activity at that moment but rather several seconds before. Given the technical limitations of fMRI (i.e. poor time resolution), we believe this was the best option for investigating the effect of baseline brain state on conscious content. Focusing on later time points would have contaminated our signal with stimulus-induced activity, thus not purely reflecting baseline brain activity.

We interpret our findings as follows: The brain state at the moment of stimulus presentation reflects baseline ongoing brain activity unaffected by the task. Our results suggest that the capacity to process external stimuli depends on the ongoing brain activity before stimulus presentation, indicating that certain connectivity configurations characteristic of conscious states facilitate external information processing.

In future studies, we aim to investigate the EEG correlates of these fMRI brain patterns. This approach will help us (1) address the issue raised by the reviewer, as the temporal resolution of EEG will allow us to pinpoint baseline brain activity closer to the stimulus onset, and (2) more easily translate our research to clinical settings, where fMRI presents significant challenges in terms of cost and transportability.

We now discuss these points in our manuscript on page 12, lines 294-308.

Minor comments:

- The rationale for including the previous stimulus detection status as a covariate is not so well explained. Please provide a bit more context in the manuscript.

The decision to include the previous stimulus detection status as a covariate was motivated by the possibility of a perceptual hysteresis effect, which refers to the influence of a preceding perception on the current one (see Chambers & Pressnitzer, 2014 for an example in the auditory domain). We hypothesized that the perception of the previous stimulus could affect the current perception, and we aimed to account for this potential effect. Indeed, we observed that previous detections increased the likelihood of current detections, demonstrating a hysteresis effect in our experiment.

Moreover, we found that brain connectivity patterns conducive to conscious perception occurred more frequently following a detection. This suggests that once the brain detects and processes a stimulus consciously, it tends to remain in or transition into states favorable to conscious perception. We speculate that the increased likelihood of participants transitioning into these favorable states after a detection might explain the hysteresis effect. Further studies are required to thoroughly explore and validate this interpretation. These points are discussed in our manuscript on page 13, lines 329-341.

It is important to note that our main results do not depend on this covariate. When we re-ran the analysis without the previous stimulus detection status as a covariate, we still found a significant pattern*SNR interaction ($\chi^2(12) = 22.27$, $p = 0.035$), with detection rates significantly higher when participants were presenting Pattern 1 compared to Pattern 2 ($t = 2.66$; $p = 0.021$, after FDR correction), Pattern 3 ($t = 2.63$; $p = 0.021$), Pattern 4 ($t = 3.42$; $p =$

0.0053) and Pattern 5 ($t = 3.28$; $p = 0.0053$) only for the threshold SNR -9. This reassures us that our results are not driven by this specific covariate and that this new model replicates our findings. We have now added this clarification to our manuscript on pages 7-8, lines 194-200.

- In the abstract, it is not so clear what “high pattern” refers to.

Following the reviewer's comment, we carefully reviewed our manuscript and noticed that the terminology in question was used only in the abstract and not in the body of the text. We thank the reviewer for bringing this to our attention. Accordingly, we have replaced the term "high patterns" with either "patterns characteristic of conscious states" or "patterns associated with conscious states" in the abstract.

- Page 11 line 261: the word “stimuli” is missing.

Thank you for bringing this to our attention. We have now corrected the sentence.

Reviewer #2:

This is an elegant study using fMRI data during rest and an auditory detection task to characterise the effects of specific brain connectivity patterns on the likelihood of consciously perceiving external stimuli. The authors provide evidence that participants are more likely to detect threshold auditory stimuli when highly connected brain patterns are present and, in turn, conscious access increases the likelihood of a subsequent highly connected pattern appearing.

The study is well conducted and the results are convincing. I only have minor comments:

We thank the reviewer for their kind words of encouragement! We are delighted that our manuscript and findings have been well received.

1. A successful detection increases the likelihood of also detecting the following stimulus and also is associated with the lingering of pattern 1. With these two results in mind, the fact that Pattern 1 is more likely to be present at the point that a new stimulus is detected could be explained by this lingering activity from the previous stimulus. In other words, pattern 1 could be associated with the brain's post-detection configuration rather than having a facilitatory effect per se. I believe this is partially addressed with the analyses displayed in Figure 4 B, that shows a reduction of the pattern occurrence after 9 seconds but I think it would be helpful to see an explicit acknowledgment of this and the authors thoughts about it in the discussion.

As the reviewer notes, our results demonstrate that (1) a successful detection increases the likelihood of detecting the upcoming stimulus and (2) there is an increased occurrence of Pattern 1 following a detection. We believe these two findings could be intrinsically linked. The first result aligns well with the concept of perceptual hysteresis, which refers to the influence of a preceding perception on the current one (Chambers & Pressnitzer, 2014). The brain's tendency to remain in or transition into favorable states following a detection could explain this hysteresis effect, facilitating the detection of the upcoming stimulus. However,

this mechanistic explanation is speculative, and further research is needed to explore this aspect in more detail.

As the reviewer points out, it is challenging to disentangle the contributions of ongoing brain activity and post-stimulus activity (induced by the stimulus and its detection) to pattern formation. It could be argued that the pattern observed at the stimulus presentation is affected by the activity induced by the previous detection. To minimize this risk, we chose unusually long inter-trial intervals (14 seconds +/- 1 second) to avoid observing the activity caused by the previous trial on the current one. Given the BOLD signal latency and duration, it is unlikely that patterns observed at the stimulus presentation are affected by the previous trial.

Another strategy we adopted to mitigate this issue was to determine our cluster centroids (patterns) using resting state data. We then labeled our task data using these cluster centroids by assigning each task fMRI volume to the nearest cluster centroid. In our approach, we did not model the BOLD response following detection in specific regions, which would be more suitable if we aimed to observe task-induced localized activity. Instead, our method reveals fluctuations in global brain states. We chose this strategy (1) to minimize the potential issue raised by the reviewer and (2) because we were interested in how ongoing brain dynamics (resting baseline brain states prior to stimulation) could influence the capacity for conscious perception.

Finally, when analyzing post-detection occurrence rates, we only considered a 9-second time window rather than the entire post-stimulus period of 14 seconds. This approach ensured that we had sufficient data points to compute post-stimulus pattern probabilities at the trial level without including patterns from the pre-stimulus period of the next trial. By doing so, we avoided using the same time points (fMRI volumes) in both analyses, which could have introduced confounding factors.

We now discuss these points on pages 11-12, lines 285-293.

2. While Pattern 1 is significantly associated with higher detection rates, several of the other patterns appear to also have detection rates above chance (this can be seen e.g. in Figure 3). I like that the authors are careful in their wording when discussing their findings and talk about 'favourable / unfavourable patterns' but I think it'd be interesting to have some reflections in the discussion about what this means for our understanding of the relationship between these patterns and conscious / unconscious states.

The reviewer rightly points out that the overall detection rates were higher than 0.5 for both Pattern 1 (one-sided Wilcoxon signed-rank test against 0.5; $V = 252$, $p = 0.002$) and Pattern 3 ($V = 203$, $p = 0.02$), while the other patterns did not statistically differ from the 50% detection performance. However, it is important to note that the overall detection rate, when all trials were considered, was not 0.5 but rather 0.58 (page 7, line 175). We aimed for a 50% detection rate for SNR-09 and therefore conducted a one-up-one-down staircase procedure prior to the experiment. Although the staircase procedure worked fairly well, resulting in a 0.58 detection rate, it did not achieve the ideal 0.50 detection rate. This discrepancy can be attributed to the experimental setting. Both the staircase and experimental procedures were conducted in the fMRI scanner during functional acquisitions.

Due to the significant scanner noise, it is very difficult to obtain a perfectly calibrated threshold in this setting.

The reviewer's comment remains valid in the sense that it is important to demonstrate that Pattern 1 specifically improved the detection rates compared to the baseline level (i.e., 0.58) to show the specificity of this pattern's facilitatory effect. Therefore, we conducted an additional analysis comparing the detection rates of each pattern to the overall detection rate resulting from our staircase procedure (0.58) using a subject-level Wilcoxon signed-rank test. As expected, the detection rate was higher than 0.58 only for Pattern 1 ($V = 231$, $p = 0.033$) and not for the others (Pattern 2: $V = 148$, $p = 0.66$; Pattern 3: $V = 174$, $p = 0.38$; Pattern 4: $V = 147$, $p = 0.66$; Pattern 5: $V = 170$, $p = 0.42$). Therefore, Pattern 1 specifically favors conscious processing, whereas the other patterns do not show such an effect.

These patterns can be interpreted as different modes of inter-areal coordination, reflecting the general brain state resulting from functional communication among various brain networks (in our study: visual, auditory, saliency, DMN, fronto-parietal, motor) at a given time point. Rather than focusing on the connectivity of a pair of regions, our method provides a global description of brain activity and how different networks interact. Our results suggest that specific modes of coordination can facilitate external information processing. For instance, the favorable pattern seems to be characterized by a decoupling of the DMN from sensory regions and the saliency network. This decoupling might allow the brain to orient its resources more towards external stimuli rather than internal information, as the DMN is primarily responsible for internally generated thoughts.

In light of this interpretation, it would be expected that brain states favorable for external information processing are expressed very rarely in vegetative state patients, who lack any signs of awareness of the external world, as previously shown by Demertzi et al. Importantly, it has also been found that these brain patterns are more present in minimally conscious state patients, who exhibit transient moments of external awareness. In the future, we aim to investigate the specific mechanisms at play (such as attention orientation) that favor such enhanced external processing by combining our current approach with localized activation analyses.

We have now incorporated these points into the manuscript on page 7, lines 184-190 and page 11, lines 278-284.

3. Please spell check the first paragraph of the discussion. There is an unfinished sentence and a typo.

Thank you for bringing this to our attention. The paragraph has now been corrected.

REVIEWERS' COMMENTS:

Reviewer #1 (Remarks to the Author):

The authors properly discuss how to interpret their findings in the discussion. They have now answered all my questions.

Reviewer #2 (Remarks to the Author):

The authors have carefully considered my comments and addressed them satisfactorily in the manuscript and rebuttal letter. I have no further suggestions and am happy to recommend their paper for publication.